# Modeling and Simulation of Si Grating Photodetector Fabricated Using MACE Method for NIR Spectrum

Akhmadi Surawijaya *, Zefanya Chandra, Muhammad Amin Sulthoni, Irman Idris and Trio Adiono

Department of Electrical Engineering, School of Electrical Engineering and Informatics, Institut Teknologi Bandung, Kota Bandung 40116, Indonesia
* Correspondence: asurawijaya@itb.ac.id

**Abstract:** In this research, we modeled a silicon-based photodetector for the NIR-IR spectrum using a grating structure fabricated using the metal-assisted chemical etching method. A nanostructure fabricated by using this method is free of defects such as unwanted sidewall metal depositions. The device is simulated using Lumerical finite difference time domain (FDTD) for optical characteristics and Lumerical CHARGE for electrical characteristics. First, we optimized the grating structure duty cycle parameter for maximum optical power absorption using the particle swarm optimization algorithm provided in Lumerical FDTD, and then used the optimized parameter for our simulations. From Lumerical FDTD simulations, we found that the Cr masker metal used in the fabrication process acts as a resonant cavity and a potential candidate for internal photo emission (IPE) effects. By using Lumerical CHARGE, we performed electrical simulation and by adding the IPE calculation we found that at 850 nm wavelength the Si grating photodetector device exhibited 19 mA/W responsivity and detectivity of $2.62 \times 10^6$ Jones for $-1$ volt operating voltage.

**Keywords:** nanophotonic; photodetectors; resonant cavity; finite difference time domain

## 1. Introduction

To realize a full silicon photonic chip, the needed photonic device must be able to be fabricated using current silicon fabrication technology. A light source such as the vertical cavity surface-emitting laser (VCSEL) has been developed to be integrated into silicon fabrication technology [1–3] and silicon-based photodetectors [4–6]. Due to silicon material limitations, silicon-based photodetectors have low responsivity to NIR communication wavelengths and so to improve their performance, many designs have been attempted such as using metal–semiconductor–metal (MSM) configurations [7,8], graphene-based configurations, [9] or using a grating structure [10–12]. Another method is to increase quantum efficiency by increasing light absorbance and photocarrier generation by using the plasmonic method [13], Schottky junction [14], and internal photo emission (IPE) effect [15,16].

In terms of fabrication, the grating structure photodetector is the simplest and, due to leaky mode resonance in the grating structure, light absorption can be enhanced further [17–19]. One of the challenges in grating photodetector structures is the sidewall surface roughness and unwanted sidewall metal deposition. This fabrication problem can be avoided by using metal-assisted chemical etching (MACE) to fabricate the grating structure; by using this method, a high-aspect-ratio grating structure can be achieved [20,21]. By using a slow-rate etching rate, sidewall roughness can be controlled to produce surface roughness around 3.6 nm [22]. The metal catalysts used in the MACE process can be used to generate IPE effects for increased responsivity of grating-type photodetectors.

In this research, we modeled a silicon grating photodetector with a sub-wavelength structure fabricated by using the MACE method with gold as the catalyst. We used chromium as the blocking layer for the MACE process and as the hard mask to etch the

grating structure. To calculate the IPE effect on the metal layer, we used IPE quantum efficiency derived by Casalino et al. [23,24]. The purpose of this research was to obtain photodetector characteristics of silicon grating fabricated by using the MACE method for NIR spectrum measurements, especially at the optic communication wavelength of 850 nm.

## 2. Materials and Methods

In this research, we used Ansys Lumerical FDTD™ for the optical simulation and Lumerical CHARGE™ for the electrical simulations. Using Lumerical FDTD, we modeled the Si grating structure and optimized it to obtain maximum optical power absorption for the 850 nm wavelength. We then calculated the photocarrier generation factor and imported it to Lumerical CHARGE. Using Lumerical CHARGE, we then performed steady-state DC analysis which also included the IPE effect from metal layers to obtain dark current and the current-voltage (IV) characteristic of the device, and calculated responsivity.

We modeled a silicon grating photodetector device shown in Figure 1 in Lumerical FDTD; the device active area was 4.25 μm by 4.25 μm, the top contact consisted of an Au-Cr layer and acted as the Schottky contact, and the bottom ohmic contact was made of aluminum. For the FDTD simulation, we set the thickness of the Au layer ($t_{Au}$) to 15 nm and the Cr layer ($t_{Cr}$) to 20 nm. For etching depth (L) and substrate thickness (D), we set them at 425 nm. The period and width of the device was 425 nm and 4.25 μm. $\Lambda$ is the grating period and a is the grating length with the duty cycle defined as $\Lambda/a$. A light source was placed on top of the structure with the plane wave propagating downward, perpendicular to the structure. Details of the MACE fabrication process are written in Appendix A. For the FDTD simulation, we defined the FDTD simulation range to cover one grating structure and set the boundary condition to periodic for the X and Y axis and PML for the Z axis.

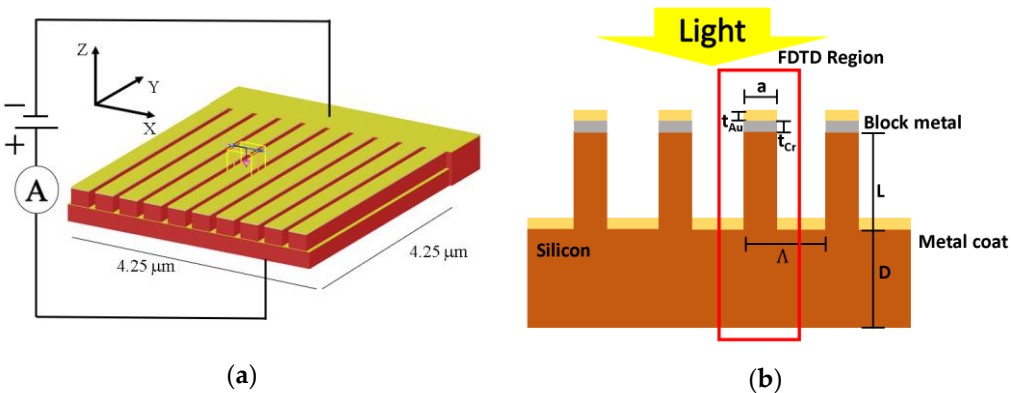

(a)                                                                                          (b)

**Figure 1.** Silicon grating photodetector fabricated using the MACE method. (**a**) Device area, (**b**) device cross-section.

FDTD simulation will calculate electric field distribution in the structure for each of the defined wavelength spectra and use it along with the material imaginary part of the complex refractive index to calculate optical power absorption per volume ($P_{abs}$). After integrating $P_{abs}$ across the structure dimension, we obtained the $P_{abs}$ total. The grating structure was defined using a custom script available in Lumerical FDTD with grating design parameters ($\Lambda$, $t_{Au}$, $t_{Cr}$, L, D, and duty cycle).

To achieve the optimum $P_{abs}$ value over the geometric design parameters, we used particle swarm optimization (PSO), available in Lumerical FDTD with duty cycle as the particle, $P_{abs}$ as the figure of merit, and criteria of 200 iterations. The PSO method flowchart is shown in Figure 2.

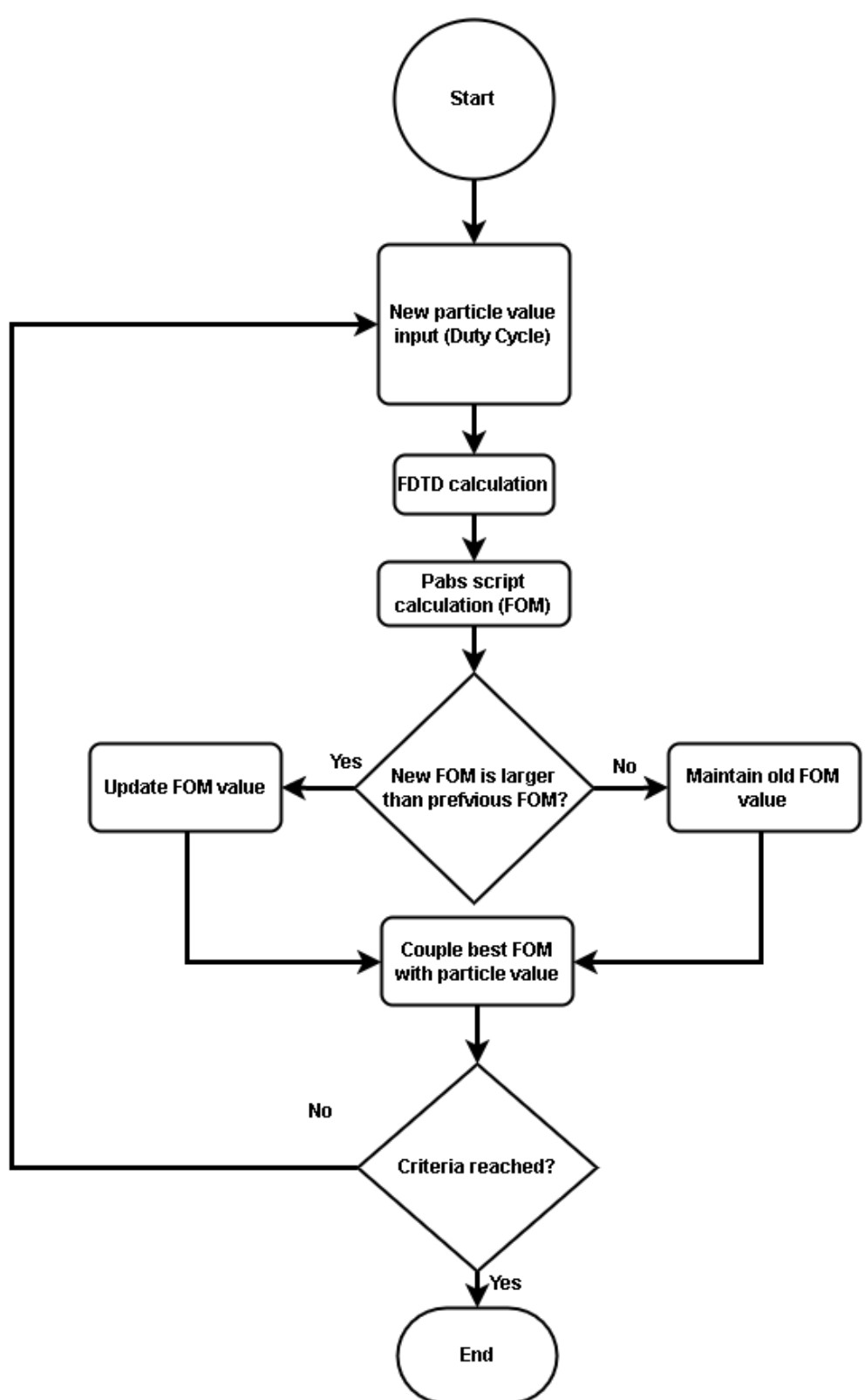

**Figure 2.** Particle swarm optimization (PSO) flowchart with optical power absorption ($P_{abs}$) as Figure of merit and duty cycle as the particle.

The optimum duty cycle value was then used as a design parameter for the rest of the simulation in Lumerical FDTD and Lumerical CHARGE. We performed FDTD with a wavelength sweep from 300 to 1600 nm to observe its optical absorption characteristic

and single wavelength simulation at 850 nm to generate photocarrier generation (G) data from the Si grating structure and optical power absorption data from the metal layers to evaluate the IPE effect probability. The maximum detectable wavelength $\lambda_m$ from metal can be calculated using Equation (1) below:

$$\lambda_m = \frac{1242}{\phi_B} \tag{1}$$

where $\phi_B$ is the Schottky barrier height which is 1.05 eV for Au-Si and 0.45 for the Cr-Si interface. The maximum detectable wavelength for the IPE process for gold is around 1182 nm and chromium is around 2760 nm, so chromium metal can be used as a light absorber up to 1550 nm for the communication wavelength. For this study, we focused on the 850 nm wavelength.

Using Lumerical CHARGE, we used the same structure model to simulate steady-state DC characteristics. For the silicon material, we set the doping level as N-type with a doping concentration of $10^{14}$ cm$^{-3}$. The top contact interface was set as the Schottky contact at the Cr-Si interface and the bottom contact was set as the ohmic contact in the Si-Al interface. G data from Lumerical FDTD were set and defined for the silicon structure. Steady-state DC characteristics were performed by sweeping the bias voltage from $-3$ to 3 volt with a 0.1 volt step, and this was performed for dark conditions (without G data) to obtain dark current characteristics and for light conditions by adding G data. The steady-state IV data were then used to calculate the device's responsivity.

To calculate the hot carrier injection from IPE, we used the quantum efficiency formula from Casalino et al. [16], as follows:

$$\eta = A_T F_e P_E \eta_c \tag{2}$$

which $A_T$ is the total optical absorptance, Fe is the Fowler factor, $Pe$ is the Vickers factor, and $\eta_c$ is the barrier collection efficiency. In Lumerical FDTD, $AT$ is the same as $P_{abs}$ total in the metal, so Equation (1) can be written in the form of responsivity as:

$$Resp = \frac{I_{IPE}}{P_{inc}} = \frac{q}{\hbar.f} \frac{P_{abs}}{P_{inc}} F_e P_E \eta_c$$

$$\frac{I_{IPE}}{P_{inc}} = q.\frac{G}{P_{inc}}.F_e P_E \eta_c$$

We then define $\eta_{IPE} = F_e.P_E.\ \eta c$ so that we obtain the following equation:

$$I_{IPE} = q.G.\eta_{IPE} \tag{3}$$

The term $G.\eta_{IPE}$ in Equation (3) is a generation scale factor that can be inserted into the Lumerical CHARGE simulation based on calculated $P_{abs}$ in metal from the Lumerical FDTD simulation. $G$ is the photocarrier generation rate data which can be calculated in Lumerical FDTD based on $P_{abs}$, as shown in Equation (4)

$$G(r) = \int \frac{P_{abs}(r,\omega)}{\hbar\omega} d\omega \tag{4}$$

## 3. Results

The PSO calculation returns a duty cycle value of 73.5% with a maximum $P_{abs}$ of 0.54. By using the 73.5% value for the duty cycle, we perform the FDTD calculation with a wavelength sweep from 300 to 1600 nm, and by using the custom Lumerical script to separate the power absorption of Si grating, Au metal contact, and the Cr metal layer we obtain $P_{abs}$ vs. lambda data, as shown in Figure 3.

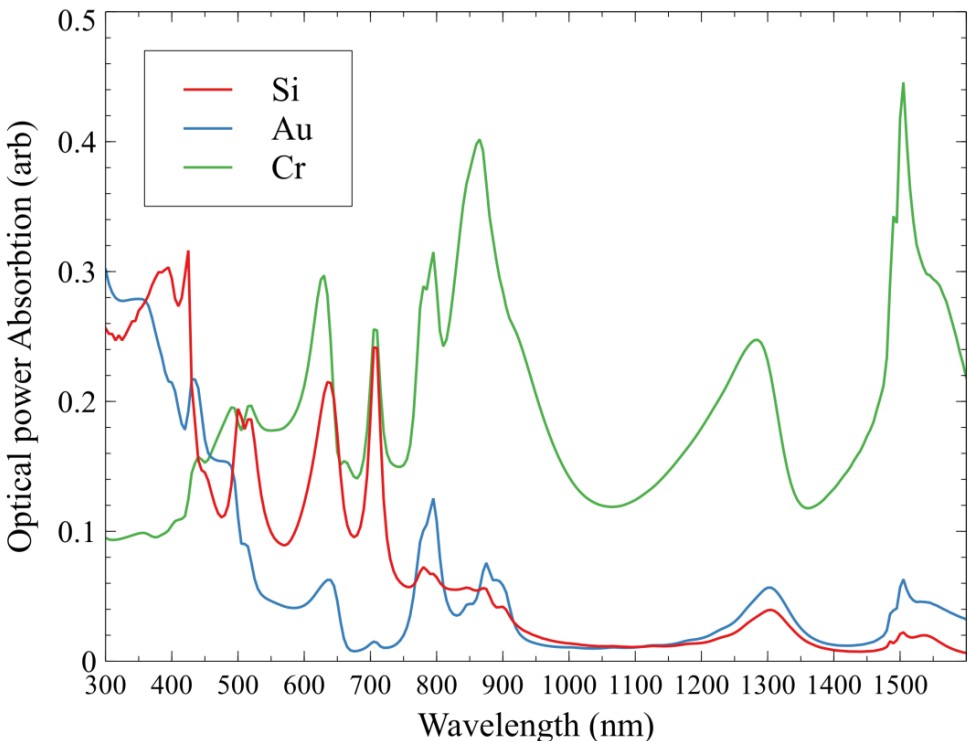

**Figure 3.** Optical power absorption spectra with a duty cycle of 73.5%. The power absorption of silicon grating, Cr metal layer, and Au metal contact is shown.

Figure 3 shows that for the NIR spectrum, silicon and gold perform rather poorly with absorbed power being less than 0.1 for the 850 nm wavelength; absorption in silicon is slightly larger compared to gold. The Cr block layer, however, shows large power absorption for the NIR and IR range; this is due to the Cr layer forming a resonant cavity layer between the Si grating and the Au metal contact. At the 850 nm wavelength, the total optical power absorbed in the Si, Au, and Cr layers is 0.055, 0.044, and 0.377, with absorption in the Cr layer being the highest. It is also interesting to see that Cr has $P_{abs}$ of 0.11 at 1050 nm and 0.41 $P_{abs}$ at 1500 nm, which means that this structure can also be used for the photodetector at these communication wavelengths.

Sharp peaks in Figure 3 are due to leaky mode resonance (LMR) in which light is absorbed into the grating structure and metal layers. Multiple peaks show multiple resonance conditions which occur in the Si grating structure, and for Si it is severely reduced for the NIR-IR spectrum while absorption peaks in the Cr layer remain strong.

A resonant cavity formed in the Cr layer results in more optical power being absorbed in the Cr. Figure 4 shows the location of the resonant cavity which is coincident with the location of the Cr layer. The location for the peak optical power absorption is in the Cr metal layer, which is sandwiched between Au and Si, and at the valley Au layer, which is used for the MACE catalyst. However, the valley Au metal layer is not directly connected to electrical electrodes; hence, it does not contribute to the overall electrical characteristics. The band diagram of the Cr/Si Schottky contacts and IPE effect mechanism is shown in Figure 5. Light which is absorbed into the Cr layer will result in the creation of hot electrons which will be injected into the Si grating structure.

To calculate the photocarrier generation rate (G) in silicon, we performed a simulation in FDTD for a single wavelength at 850 nm with an optical power source of 1 watt normalized to the device area, which was 425 μm × 425 μm. G data were then exported into Lumerical Charge for the steady-state DC calculation. We also set the recombination rate in silicon material to include the Shockley-Read-Hall recombination, Auger recombination, and radiative recombination.

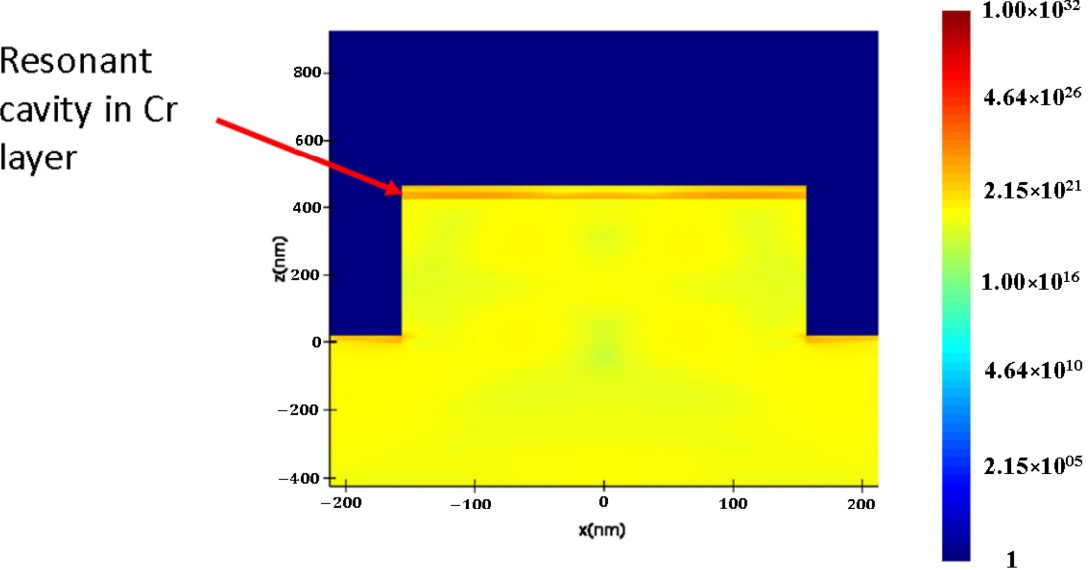

**Figure 4.** Distribution of optical power absorption ($P_{abs}$) in the Si grating structure at 850 nm wavelength. A resonant cavity forms in the Cr layer for the 850 nm wavelength which is located between the Au top contact and the top Si grating surface (shown by red arrow).

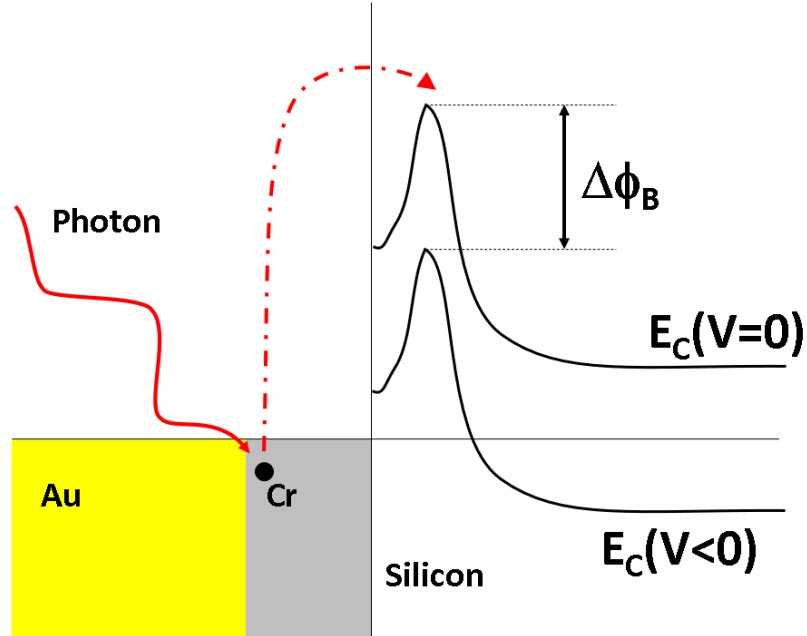

**Figure 5.** Band diagram of Schottky contact and IPE effect mechanism.

Using Lumerical CHARGE, we used a custom Lumerical script to set the simulation for dark current calculation by setting G data off and setting the light current by using the G data, scaled by a factor to simulate various optical power. Here, we set the optical power at 1 watt, 500 mW, and 250 mW simply by setting the G data scale factor in the Lumerical script. For the steady-state DC simulation, the IPE factor was not considered. The resulting I-V characteristic and responsivity are shown in Figure 6.

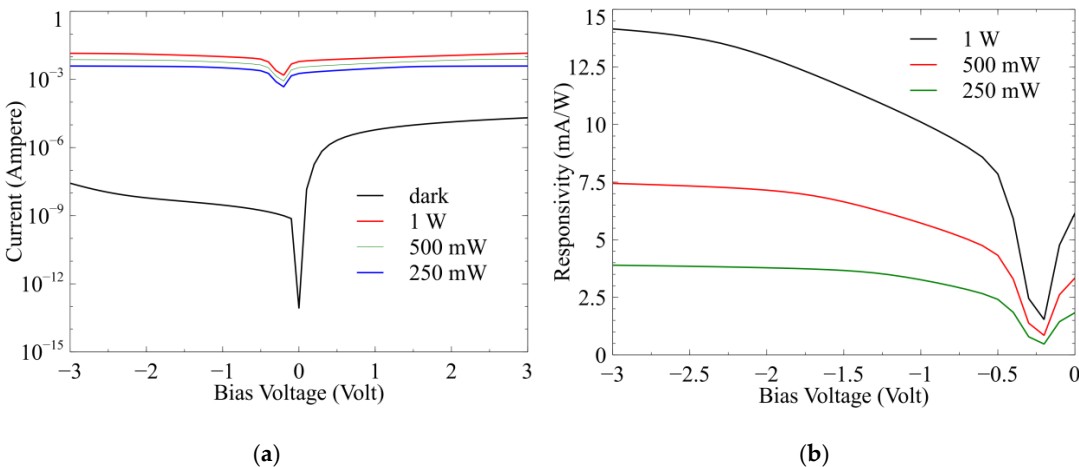

**Figure 6.** Electrical characteristic of Si grating structure at 850 nm wavelength. (**a**) Steady-state DC characteristic for dark current and light current with optical source power at 1 watt, 500 mW, and 250 mW. (**b**) Responsivity for reverse bias condition with varying optical source power at 1 watt, 500 mW, and 250 mW.

From Figure 3, we can see the photovoltaic characteristics at zero bias in which a photocurrent of 6.1 mA is flowing in the device for 1 watt of optical power. For an operating voltage of −1 volt, the lowest dark current is 2.9 nA, and the maximum light current is 10.1 mA, 5.7 mA, and 3.2 mA for optical powers of 1 watt, 500 mW, and 250 mW, respectively. By using the DC steady-state data, we can calculate device responsivity by using the formula R = ($I_{light}$ − $I_{dark}$)/source power, and the result is shown in Figure 6b.

For the IPE efficiency for the chromium metal layer, we set chromium $\phi_m$ = 4.6 eV, and $L_e$ = 19 nm. Since the IPE effect is only valid for reverse bias conditions (forward bias will result in zero efficiencies), we set the sweep voltage as the calculation from 0 to −3 volt, and the result is shown in Figure 7. The IPE effect efficiency increases as the reverse voltage increases; this is because of the barrier-lowering effect on the Schottky barrier between chromium and silicon due to the mirror charge in the chromium–silicon interface.

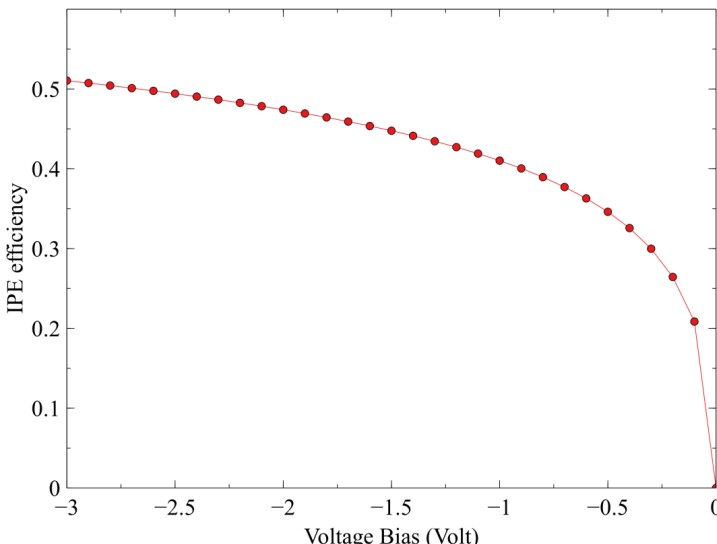

**Figure 7.** IPE effect efficiency ($\eta_{IPE}$) for 20 nm chromium thin film for 850 nm light source.

By using the $\eta_{IPE}$ value from Figure 6, we can calculate the IV characteristic with total photocurrent as a combination of the photogenerated carrier from silicon material and hot carrier injection from the IPE effect in the chromium metal layer. The result is

shown in Figure 8, which also compares the light current with IPE and without IPE effect. It is shown that IPE contribution became larger as the reverse current increased, which is directly related to the barrier lowering in the metal–Si interface.

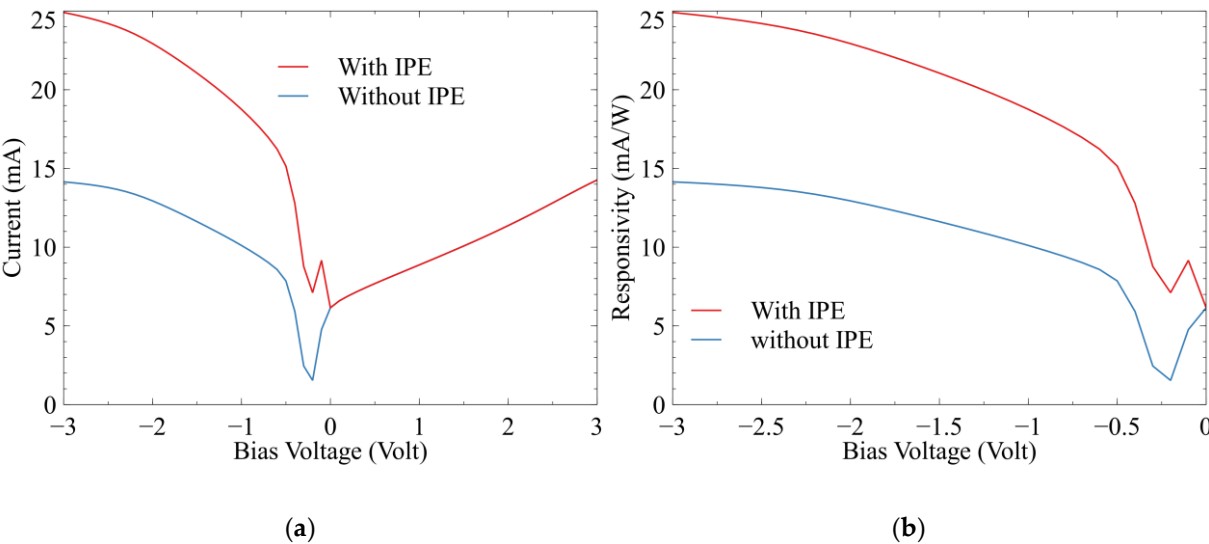

(**a**)　　　　　　　　　　　　　　　　　　　　　　　　(**b**)

**Figure 8.** Electrical characteristic with IPE effect included. (**a**) IV characteristic of the device with light current for photocurrent without IPE and photocurrent with IPE, (**b**) responsivity comparison with and without IPE effect.

The IPE effect from the absorbing metal layer can increase the silicon photodetector device's responsivity by injecting a hot carrier and it is very beneficial due to silicon's low absorptivity in the NIR light spectrum. The responsivity comparison is shown in Figure 8b and the increase in responsivity due to the IPE effect is around 86%.

## 4. Discussion

Here, we have demonstrated that by using the MACE process to fabricate a silicon grating photodetector, we can use the MACE block metal as a resonant cavity to absorb light which will induce the IPE effect to generate hot electrons and inject into the silicon grating structure. The gold layer used in the MACE etching can be left in the bottom of the grating structure and act as a reflector to allow light to be absorbed in the grating structure. To evaluate the performance of our novel MACE-fabricated Si grating photodetector, we compared it with similar 850 nm photodetectors, as shown in Table 1.

**Table 1.** 850 nm photodetectors' performance comparison.

|  | Device Area ($m^2$) | Operating Voltage (Volt) | Dark Current (A) | Responsivity (A/W) | Quantum Efficiency | Detectivity (Jones) |
|---|---|---|---|---|---|---|
| Feng [12] | $6.4 \times 10^{-9}$ | −2.0 | $2.9 \times 10^{-9}$ | 0.386 | 0.56 | $3.21 \times 10^8$ |
| Diels [25] | $3.07 \times 10^{-9}$ | −1.5 | 40 | 0.00027 | 0.00039 | 4.18 |
| Fard [4] | $8.4 \times 10^{-11}$ | −12.0 | 0.4 | 0.15 | 0.22 | $3.84 \times 10^3$ |
| Youn [26] | $10^{-10}$ | −10.2 | 100 | 4.67 | 6.81 | $8.26 \times 10^3$ |
| This work | $1.83 \times 10^{-11}$ | −1.0 | $2.94 \times 10^{-9}$ | 0.019 | 0.03 | $2.62 \times 10^6$ |

Our proposed device has the smallest device area which resulted in lower responsivity, but the detectivity value is nearly comparable to a larger device. Using the Schottky contact can greatly reduce the dark current and, at the same time, the metal layer can be set to provide the hot carrier with the IPE process. In our proposed device, the chromium layer

serves two purposes: first, as the blocking layer for the MACE process, and second as a resonant cavity for the light absorber and IPE source.

Our study has revealed that using the MACE process to fabricate Si grating for the NIR-IR photodetector will not only prevent defects such as unwanted sidewall metal deposition but also can enhance optical power absorption in the NIR-IR spectrum region, especially in the metal mask used in the MACE process since it can act as a resonant cavity.

## 5. Conclusions

We have modeled and simulated a silicon grating photodetector fabricated using the MACE process and found that the Cr masking layer acts as a resonant cavity for optical power absorption, which became the source for the IPE effect. By including the IPE effect in the Cr layer, we can achieve a responsivity of 19 mA/W with detectivity of $2.62 \times 10^6$ Jones for light detection at 850 nm.

**Author Contributions:** Data curation, A.S. and Z.C.; methodology, A.S.; project administration, A.S.; supervision, I.I. and T.A.; validation, M.A.S.; writing—original draft, A.S. All authors have read and agreed to the published version of the manuscript.

**Funding:** This research received no external funding.

**Acknowledgments:** We would like to thank the Research Center for Nanosciences and Nanotechnology (RCNN), Institut Teknologi Bandung, for providing us with Lumerical simulation tools for our research group.

**Conflicts of Interest:** The authors declare no conflict of interest.

## Appendix A

In this section, we will briefly explain the proposed fabrication method by using MACE. An illustration of the fabrication process is shown in Figure A1 below. For the device fabrication, we used an N-type silicon wafer with a doping concentration of $10^{14}$ cm$^{-3}$. A 5 nm Cr metal mask for the MACE process was fabricated on top of the substrate by using the lift-off method (a). An Au layer (15 nm) was then deposited on the surface to act as the MACE catalyst (b). The sample was then inserted into a mixture solution of $H_2O_2$ and HF (1:1) and a grating structure was formed through the MACE process (c). After forming the grating structure, the Al ohmic contact was deposited on the backside of the Si substrate.

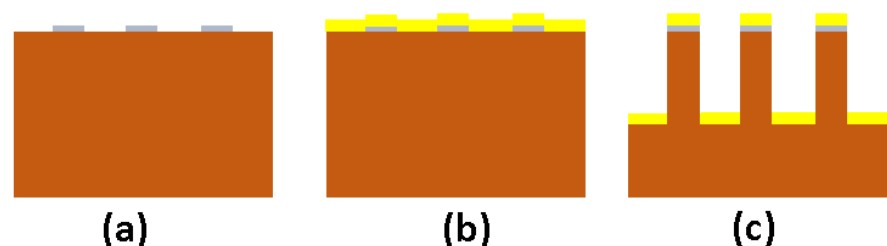

**Figure A1.** Grating structure fabrication process using MACE method.

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
