# Peer review of "Modeling and Simulation of Si Grating Photodetector Fabricated Using MACE Method for NIR Spectrum"

_electronics, doi:10.3390/electronics12030663_

Round 1
Reviewer 1 Report
Integrating light sources and high-efficiency Photodetectors are currently the most important technical challenges Photonics Integrated Circuits are facing. Bringing light generation and detection on the chip can explode the significance of PICs in many applications. Therefore, the results presented in this paper have both scientific and industrial importance. The simulations are done to validate the design and the results are presented very well.
Below are a few important comments, suggestions, and changes authors should consider before this paper is published:
1. Abstract: Authors should not use abbreviations directly in the abstract. For example in lines 11 and 17 FDTD and IPE should be defined
2. Define MSM in line 28 and similarly, authors should pay attention to defining abbreviations in the paper. Line 53 ”IV” is not defined
3. Line 31: Internal Photo Emission should have small first letters. In addition, IPE is an effect, not a method. The context in which IPE is used in this sentence does not fit. Please mention the methods which produce the IPE effect in reference 15,16
4. Line 41: “the gold” and “the Chromium”
5. Line 59: the symbol of substrate thickness is the capital letter “D” not the small letter “d” as shown in Figure 1
6. Authors should add a subscript to all the symbols in this paper for example in line 107 it should be AT and in line 108 it should Pabs. Please correct this in the entire paper
7. Define the symbol “G” used in Equation (3)
8. All Equations should be in the center.
9. Figure 3: The authors should explain this Figure more clearly. In line 126 it says the Figure represents absorption for each of the grating materials which means each of these materials is used for grating instead of Si. While in lines 131-132, it has been said that the Cr layer is forming a cavity between Si grating and the Au metal contact. These statements seem contradictory. The authors should clarify this in the text.
10. Is this possible for authors to explain the origin of sharp peaks and troughs for Si, Au, and Cr in Figure 3
11. Can authors CHARGE Boundary conditions
12. Line 179: “it is very beneficial”
13. The authors emphasized that the MACE process helps prevent sidewall roughness but this has not been explained in the main text how it works.
14. Figures 3,5,6,7,8,9 should be considered to be places in column format of (1,2) to make them more presentable.
15. References should be checked. Now for example “nm” is written as “Nm”.
Overall authors should polish the text and improve some minor yet important text mistakes.
Some general comments; more for my understanding purpose rather than suggestions.
i) Can authors somehow compare their work with references 11, 12 where authors report much higher responsivity?
ii) I am also curious to know the origin of the peak in the I/V curve for IPE in Figures 8 and 9. Is this possible for the authors to give an explanation in the response letter? See Figure below

Reviewer 2 Report
In this paper, Surawijaya et al. simulated the Silicon-based photodetector of nanocolumn using Lumerical FDTD for optical characteristics and Lumerical CHARGE for electrical characteristics. The manuscript is publishable with suggestions noted below:
1. The author states in the abstract and intro that the "Metal Assisted Chemical Etching method is free of defects such as sidewall roughness and unwanted sidewall metal depositions." This is not true because sidewall roughness is often a big problem for MACE. It is also related to the level of roughness the author is referring to. For e.g. 100 nm roughness is big for nanowires, while may not be significant for microwires.
2. Figure 1b should be drawn scaled to the dimension. L and D should be the same length but looks quite different.
3. "substrate thickness (d)", should be "substrate thickness (D)".
4 ". Here " should be "Here".
5. The band diagram of the Schottky junction should be provided.
6. Fig. 4 is confusing. Why is the y-axis not normalized for comparison? If I understand correctly, the grating materials are bare Si, Si/Au, Si/Cr. It should be corrected in the figure caption.
Round 2
Reviewer 2 Report
My concerns have been addressed, and the article should be publishable.